# First Record of the Velvet Ant *Mutilla europaea* (Hymenoptera: Mutillidae) Parasitizing the Bumblebee *Bombus breviceps* (Hymenoptera: Apidae)

**DOI:** 10.3390/insects10040104

**Published:** 2019-04-12

**Authors:** Wenting Su, Cheng Liang, Guiling Ding, Yusuo Jiang, Jiaxing Huang, Jie Wu

**Affiliations:** 1College of Animal Science and Veterinary Medicine, Shanxi Agricultural University, Taigu 030801, Shanxi, China; 18404983218@163.com (W.S.); jiangys-001@163.com (Y.J.); 2Sericulture & Apicultural Research Institute, Yunnan Academy of Agricultural Sciences, Mengzi 661101, Yunnan, China; liang1087@163.com; 3Key Laboratory for Insect-Pollinator Biology of the Ministry of Agriculture and Rural Affairs, Institute of Apicultural Research, Chinese Academy of Agricultural Sciences, Beijing 100093, China; dingguiling@caas.cn (G.D.); apis@vip.sina.com (J.W.)

**Keywords:** Bumblebee, *Bombus breviceps*, velvet ant, *Mutilla europaea*, parasitoids

## Abstract

Mutillid wasps are ectoparasitic insects that parasitize the enclosed developmental stages of their hosts. Adults are sexually dimorphic, with brilliantly colored and hardened cuticles. The biology of parasitic mutillid wasps has rarely been addressed. Here, we investigated the parasitization by *Mutilla europaea* on an important pollinator, *Bombus breviceps*. The parasitic biology and dispersal ability of *M. europaea* were observed and tested under experimental conditions. We provide the first record of *M. europaea* parasitizing *B. breviceps* in southwestern China. As is the case with other bumblebee species, *M. europaea* mainly parasitized the puparia of males. The dispersal and invasion ability of this parasite under experimental conditions indicates that it spreads rapidly, as far as 20 m in one week, and invades different hosts (*B. breviceps* and *Bombus haemorrhoidalis*). This report not only clarifies the parasitic relationship between *M. europaea* and *B. breviceps*, but also has important ecological implications for the conservation of bumblebees in China.

## 1. Introduction

Most velvet ants, *Mutilla europaea* (Hymenoptera: Mutillidae), are solitary parasitoid wasps characterized by strong sexual dimorphism, with females being apterous and males being winged. *Mutilla* is a genus of the Mutillidae family [1]. When they are not actively feeding and enclosed in some protective structure (e.g., cell, cocoon, puparium, ootheca), *Mutilla* species are larval ectoparasitoids of the developmental stages of other insects, mostly other aculeate Hymenoptera [2,3]. The known hosts of velvet ants include a wide variety of wasps and bees in the order Hymenoptera [2,3,4] and some species of Diptera [5,6], Coleoptera [7], Lepidoptera [8], and Blattodea [9]. Despite its size (with more than 4300 described species), few studies have addressed the biology of the family Mutillidae [10,11]. Recently, it was estimated that only approximately 2–3% of parasitic hosts have been identified. In addition, only a few velvet ant species have been recorded to parasitize social insects, such as ants, honeybees, and bumblebees [3].

Bumblebees, *Bombus breviceps* (Hymenoptera: Apidae), are important pollinators that are distributed in almost all terrestrial habitats and play an important role in agricultural production and ecosystem balance [12,13]. As primitively eusocial insects, bumblebees generally nest on or under the ground [14]. The first description of *Mutilla europaea* L. within a nest of *Bombus muscorum* L. was provided by Christ in 1791 [15], but the parasitic relationship between bumblebees and *M. europaea* was not confirmed until the 1840s [4]. Further studies have indicated that several mutillid species use bumblebees as hosts, and they can attack numerous bumblebee species [2,3]. Because of its diverse landforms and rich vegetation, China harbors the highest variety of bumblebee species in the world [16,17,18], providing the opportunity to study the basic biology, especially the wide host selection, of mutillids.

*Bombus breviceps* belongs to the subgenus *Alpigenobombus*, and it is widespread throughout Asia [17]. Deka’s studies demonstrated that *B. breviceps* is the main pollinator of greater cardamom (*Amomum subulatum* Roxb.) in Sikkim [19,20]. In recent years, we have reared native bumblebee species and have selected some species with potential commercial applications (*Bombus terrestris*, *Bombus lantschouensis*, *Bombus patagiatus*, *Bombus ignitus* [21], and *B. breviceps*, unpublished data). Such rearing of native bumblebees makes it possible to study the parasitic biology of mutillids.

In this study, we aimed to classify the parasitic biology of *M. europaea*. We first reported that *M. europaea* parasitizes the Asian bumblebee species *B. breviceps* in the field. Furthermore, we documented the biology of *M. europaea* in *B. breviceps* colonies maintained in the laboratory and recorded the dispersal ability of this parasite in a confined environment.

## 2. Materials and Methods

### 2.1. Samples

Individuals of the native bumblebee species *B. breviceps* and *Bombus haemorrhoidalis* were reared from wild queens collected in the early spring of 2016. The queens (60 *B. breviceps* and 8 *B. haemorrhoidalis*) were individually reared in small wooden boxes (28 cm × 21 cm × 14 cm) in the laboratory (in darkness at 29 ± 1 ℃, 57 ± 2% relative humidity) for colony development. They were fed ad libitum with sugar syrup (50% sugar content, w/w) and commercially available pollen collected by *Apis mellifera*. A total of 10 colonies of *B. breviceps* and two colonies of *B. haemorrhoidalis* with similar sizes were selected for the dispersal experiment. The colonies were artificially reared from a single queen under laboratory conditions. Therefore, the colonies were not parasitized by velvet ants in the following dispersal ability experiment.

Bumblebee nests are difficult to find in the field, because they occur underground or in dense vegetation [22,23]. Therefore, we located bumblebee colonies via consultations with local farmers and by tracking the workers. From 4–11 September 2016, we found six naturally occurring nests of *B. breviceps* in an area of 3 km^2^ near the village of Shidongba (E103.0705; N23.4006), located in the south of Yunnan Province, China (Figure 1A–C). The nests were removed by digging and were immediately placed in the prepared wooden breeding boxes. All these colonies were checked for the occurrence of velvet ants and were transported to the laboratory for rearing.

### 2.2. Biology of B. breviceps Parasitism by M. europaea

Wild *Bombus breviceps* colonies were kept in dark rooms at a temperature of 29 ± 1 ℃ and with relative humidity of 57 ± 2 % and were fed sugar syrup and pollen as stated above. Every two days, we randomly selected and dissected eight bumblebee cocoons in each colony under red light to check for the presence of velvet ants. We also observed the basic behaviors of *M. europaea* in bumblebee nests, such as feeding, copulating, spawning, and aggression. Six bumblebee cocoons containing *M. europaea* eggs were separately incubated (28 ± 1 ℃) to estimate the developmental time of velvet ants.

### 2.3. Observation of the Dispersal Ability of M. europaea

After 45 days, at the end of bumblebee colony development, five wild colonies infected with *M. europaea* were placed in the innermost area of a closed yard (90 m × 30 m), and two nonparasitized *B. breviceps* colonies were placed in each position located at distances of 1 m, 5 m, 15 m, 20 m, and 80 m from the *M. europaea*-parasitized colonies. In addition, two colonies of *B. haemorrhoidalis* (a common local bumblebee species) without *M. europaea* were placed at a distance of 10 m to evaluate whether interspecies infection occurred. All the colonies were arranged in a straight line according to distance and were fed syrup and pollen as stated above. Moreover, the bumblebees were free to forage on wild flowers. We checked the colonies for *M. europaea* parasitism every seven days and recorded the date when a female velvet ant was present. After a female was observed in a nest, cocoons were randomly selected and dissected with scissors within one week to confirm whether the colonies were parasitized. Parasitism was considered to have occurred when velvet ant larvae were found in the dissected cocoons. These cocoons were damaged in the process of the observation.

## 3. Results

### 3.1. Field Investigation of the Parasite

Bumblebee species other than *B. breviceps* have been recorded in this area before [18]. However, we did not find any nests of other species during the field investigation. Five of six collected *B. breviceps* nests were located in bushes (Figure 1A,C) that were surrounded by a thick, dense layer of grass and were located approximately 5 cm above the root collar of the bush. The remaining nest was located in a rat cave, which was approximately 10 cm below ground. The naturally occurring colonies were of variable size, with each colony containing a population of 30–200 workers, as well as pupae (Table 1). We observed adult females and males of *M. europaea* living in five of the six nests (Table 1, Figure 2); velvet ants were not found in the smallest colony (30 workers). The queens and workers in the parasitized colonies seemed to develop normally.

### 3.2. Biology of M. europaea Parasitism in B. breviceps Colonies

All the natural *B. breviceps* nests reared under laboratory conditions contained female and male *M. europaea* moving freely within them without attack from the bumblebees. Female *M. europaea* were observed feeding on the pollen and sugar syrup supplied to the bumblebees. A small number of adult *M. europaea* occasionally crawled out of their original host nests and moved about in the dark room. Sometimes a male was observed lying on the back of a female, showing behavior that was similar to copulation behavior.

We observed that the bumblebees were parasitized during the prepupal stage, when they had stopped feeding and their cocoons were being spun or the puparia had just formed. Female *M. europaea* penetrated the wax wall and laid eggs on the surface of the *B. breviceps* puparium (Figure 3A). We found that more than one egg but fewer than four eggs were laid on a single puparium but that only one egg successfully hatched and developed into a larva (Figure 3B). The larvae of *M. europaea* laid on the pupa and sucked the body fluid of the *B. breviceps* pupa (Figure 3C,D). In addition, double cocoons could be observed when the cocoons of *B. breviceps* were dissected (Figure 3E). As the larvae of *M. europaea* developed, they sucked out the body fluid of the *B. breviceps* pupae, and these bumblebee pupae eventually died.

The eggs of *M. europaea* hatched in three days, and the total developmental time from egg to imago was approximately 26 ± 1 days, while other development stages were not recorded in detail. We were able to tell the difference between female and male *M. europaea* at the pupal stage, as the female ovipositor and the male genitalia could be observed (Figure 3F). At the end of colony development (colonies 1–5), we removed the complete brood and dissected all the bumblebee brood cocoons to estimate the percentage of *M. europaea* parasitism. All male pupae were parasitized. In addition, approximately 50% of worker pupae were parasitized, which were similar in size to the male cocoons. We did not find *M. europaea* larvae in the queen pupae in any of the colonies.

### 3.3. Dispersal Ability

When the colonies were placed in the closed yard one week later, *M. europaea* females were observed within the mutillid-free colonies less than 20 m away from the originally parasitized colonies. Moreover, we observed female *M. europaea* wandering around the entrance of the mutillid-free colonies and freely entering the colonies. Ten bumblebee colonies within 20 m were parasitized after one week, including the two *B. haemorrhoidalis* colonies at a distance of 10 m and eight *B. breviceps* colonies at distances of 1 m, 5 m, 15 m, and 20 m. The two *B. breviceps* colonies at 80 m were invaded by *M. europaea* on the 21st day. We sampled some cocoons randomly from each test colony and found that all colonies were confirmed to contain *M. europaea* larvae one week after a female *M. europaea* entered the nest.

## 4. Discussion

It has been reported that *M. europaea* can use different bumblebee species as hosts [3,24]. In this study, we confirmed that *M. europaea* can parasitize both *B. breviceps* and *B. haemorrhoidalis*, which further extends the host range of this mutillid. Moreover, we observed that no apparent barriers needed to be overcome by *M. europaea* shifting to the different host species. In Yunnan Province, *B. breviceps* and *B. haemorrhoidalis* are bumblebees with high abundance and sympatric distributions in the wild [25]. According to our investigation, *M. europaea* are frequently parasitizing in natural *B. breviceps* nests. This suggests that we should pay attention to the population dynamic changes of this bumblebee for conservation, because a large number of bumblebee individuals may be parasitized at once, which inflicts high costs on bumblebee colonies. In addition, the extensive bumblebee diversity in the area also requires our attention, as *M. europaea* can use different bumblebee species as hosts [4]. The ability to parasitize multiple species would at least compensate for the apterous form of the females of this mutillid, which limits its range of distribution to some extent. Further investigation is required to determine whether there are any preferences in terms of the available hosts.

We found that the *M. europaea* females easily entered the colonies of *B. breviceps*; they even freely moved around the nests and were not attacked by workers. This observation is consistent with that reported by Hoffer [24]. In addition, this behavior was also recorded in another *M. europaea* host, the social wasp *Polistes biglumis* [26]. The immunity of the velvet ant to the host may be attributed to its heavily sclerotized cuticle and the venomous stings it imparts to the host [27]. Chemical signals from the velvet ant may also play an important role. It has been reported that the epicuticular lipids of velvet ants contain fewer recognition cues than their *P. biglumis* hosts, which may allow velvet ants to visit wasp nests without being significantly attacked [28]. However, *P. biglumis* has been reported to fight *M. europaea* [29], and there are reports of *Bombus ignitus* removing female *M. europaea* from the nest [30]. Therefore, the aggression of the host against the velvet ant may be affected by the status of the host, such as the colony size or developmental stages present [30].

According to our investigation, male pupae were 100% parasitized by *M. europaea*. In addition, the low rate or lack of parasitism among workers (approximately 50%) and queens (0%), respectively, peaked our interest. The reasons for the high parasitism rate among drones may be as follows: First, compared with workers, drone puparia may provide many more nutrients or special components that attract female velvet ants to lay eggs [31]. Second, velvet ants are ectoparasitic insects. To protect the hosts for subsequent generations, drones may be selected for parasitization, because the number of drones is higher than that of the queens and workers, especially at the end of colony development. However, we need to conduct additional experiments to test these speculations.

In the field, five of the six *B. breviceps* colonies had been invaded by *M. europaea*. This high parasitism rate in the field indicates that the dispersal ability of *M. europaea* is greater than our expectation, even though the females is apterous. The test of the dispersal ability in the laboratory also demonstrated that *M. europaea* is able to disperse at a relative high speed (about 80 m/20 days). In some hymenopteran insects, the winged males help to carry the females to expand the area of invasion [32]. However, we did not observe this phenomenon in our experiment. We observed that female *M. europaea* crawled freely about the yard and sought new bumblebee nests. It has been reported that the fecundity of velvet ants is higher than that of bumblebees [3,4]. Therefore, the combination of the high dispersal ability and fecundity of velvet ants poses a serious threat to local bumblebees. Base on the parasitizing rate of *B. breviceps* in our experiment, the number of males will decrease significantly when the colonies are parasitized, and this will lead to a mating problem, such that virgin queens may not mate with high-quality males.

Our study presents a new host bumblebee, *B. breviceps*, of *M. europaea* in southwestern China. *M. europaea* is mainly parasitic on the pupae of males. The dispersal experiment showed that *M. europaea* spreads rapidly and invades different hosts (*B. breviceps* and *B. haemorrhoidalis*) under experimental conditions. This report not only indicates a parasitic relationship between *M. europaea* and *B. breviceps*, but also has important ecological implications for the conservation of bumblebees in China. However, how bumblebees protect themselves against the invasion of velvet ants remains unclear. Therefore, research on effective management to protect bumblebees from this social parasite needs to be conducted in the future.

## 5. Conclusions

In conclusion, we first record a velvet ant, *M. europaea* parasite on an oriental bumblebee, *B. breviceps* in southwestern China. *M. europaea* is mainly parasitic on the pupae of males. Under the experimental conditions, *M. europaea* invades different host bumblebees, *B. breviceps* and *B. haemorrhoidalis*. Furthermore, we demonstrated that *M. europaea* has a high dispersal ability in a long-distance host. More studies are needed to explore how bumblebee protect themselves to *M. europaea*. Our research on the parasitic biology of *M. europaea* in bumblebee, *B. breviceps*, can contribute to the conservation of bumblebees.

## Figures and Tables

**Figure 1 insects-10-00104-f001:**
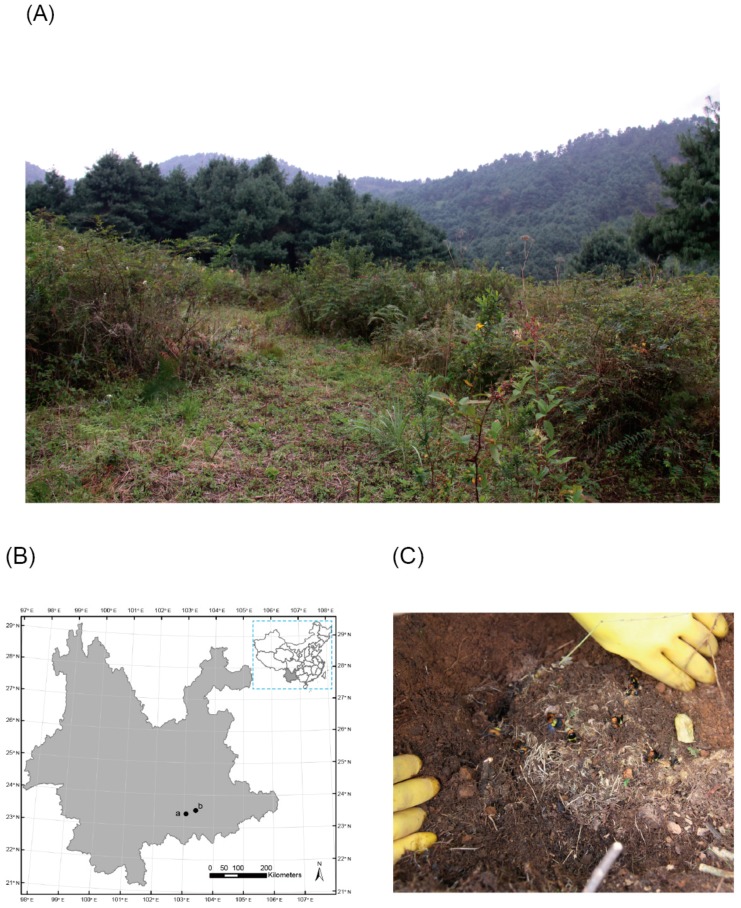
Sampling and experimental site information. (**A**) Habitat of the first bumblebee colony collected; (**B**) points represent the sampling and experiment GPS (Global Positioning System) coordinates: point a is the collection site and point b is the dispersal experiment site; (**C**) *Bombus breviceps* colony in the field.

**Figure 2 insects-10-00104-f002:**
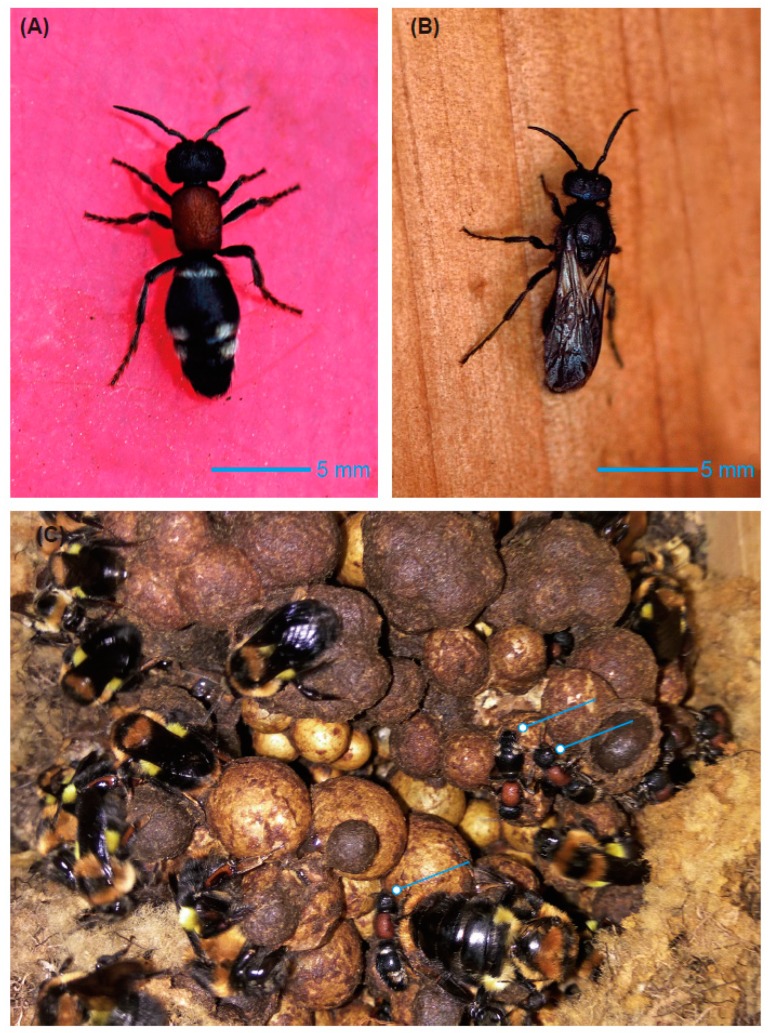
Adult *Mutilla europaea*. (**A**) Female *M. europaea*; (**B**) male *M. europaea*; and (**C**) female *M. europaea* inside a *B. breviceps* nest.

**Figure 3 insects-10-00104-f003:**
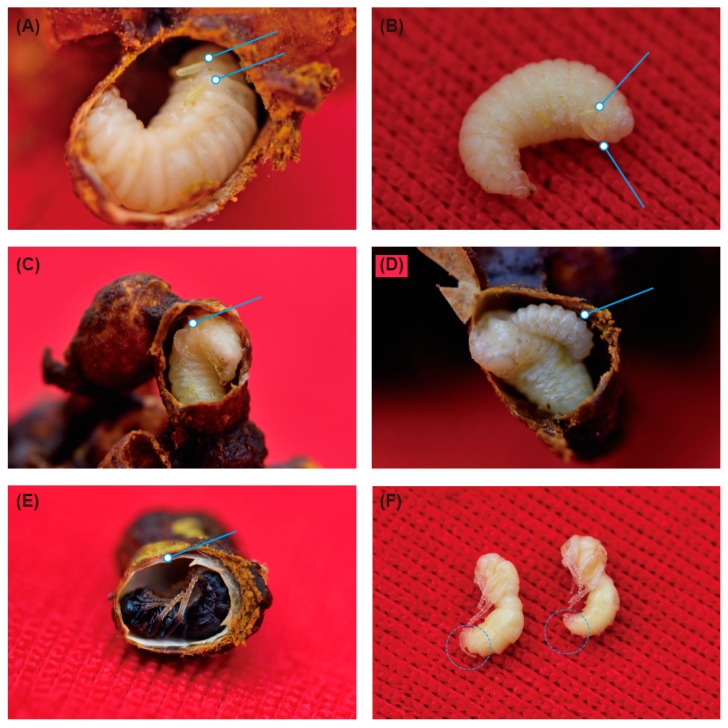
Parasitic biology of *M. europaea* in a *B. breviceps* colony. (**A**) Two eggs in a single cell of *B. breviceps*; (**B**) one hatched egg (downward arrow) and one depauperate egg (upward arrow) in a single cell of a *B. breviceps* pupa; (**C**) young larva of *M. europaea* sucking the body fluid of a *B. breviceps* pupa; (**D**) developing *M. europaea* larva and collapsing appendages of the *B. breviceps* pupa; (**E**) double cocoons; (**F**) white pupae of *M. europaea*, the dotted circles show the ovipositor of the female (**left**) and the genitalia of the male (**right**)**.**

**Table 1 insects-10-00104-t001:** Bumblebee *B. breviceps* and velvet ant *M. europaea* stages in each of the field colonies.

Colony ID	Queen	Virgin Queen	Worker	Queen Pupae	Female of *M. europaea*
1	Yes	0	80	15	20
2	Yes	0	200	13	>10
3	Yes	0	150	10	>10
4	Yes	10	150	20	>10
5	Yes	0	200	20	>10
6	Yes	0	30	0	0

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
