# Peer review of "First Record of the Velvet Ant *Mutilla europaea* (Hymenoptera: Mutillidae) Parasitizing the Bumblebee *Bombus breviceps* (Hymenoptera: Apidae)"

_insects, 2019, doi:10.3390/insects10040104_

Round 1

Reviewer 1 Report

The report titled "First record of the velvet an Mutilla eropaea----" contains some novel finding about the biology of the velvet ant in China, this draft could be submitted from this journal with minor revision. My minor comments are as follows.

Page2 line 64.  We can not understand how many colonies were propagated from queens and used for field test of parasitization. I guess that the numbers were 10. The authors should state more clearly.

Page 3 line 100. One nest -> The remaining one nest

Page 3 line 121. these pupae were unable to develop normally -> this means that these pupae died?

Page 3 line 126-128. The authors should insert the following sentence, like, "even in the presence of queen pupae,"

Page 3 line 133. Why "unfortunately"?

Page 3 Dispersal ability. The authors should explain how many colonies were infested by the velvet ants in more detail. Eight intact colonies within 20 m were infested from the velvet ants within one week ?

Figure 3. We could not see well the eggs and hatched eggs due to overlapping the arrows in particular Figure A to C.

Table 3. Insert the caption for this table. I did not understand "Virgin" in this table.

Page 7 line 176-178. If the size of pupae is important for parasitization, queen pupae should be used. However, the data is contradicted with this trend.

Page 7 line 178-181. ectoparasitoids -> do not use bald letter. I do not understand the meaning of "velvet ants are ectoparasitoids". I can not understand the following sentences.

Page 7 line 196-198. I can not understand the meaning of the two sentences. The authors should modify the last sentence of the paper. This part is one of the most important parts of the paper. 

Reviewer 2 Report

The manuscript presents a novel host association for velvet ants and bumblebees. There are serious flaws, however, with how the results are presented and the soundness of the discussion. The main flaw is that results are presented in a descriptive way, therefore making the article suitable for a more local, lower impact journal. There are no quantitative results and the discussion is highly speculative.

The english language style should also be extensively corrected. 

More detailed comments in the attached file. 

Author Response

Reviewer # 2

The manuscript presents a novel host association for velvet ants and bumblebees. There are serious flaws, however, with how the results are presented and the soundness of the discussion. The main flaw is that results are presented in a descriptive way, therefore making the article suitable for a more local, lower impact journal. There are no quantitative results and the discussion is highly speculative.

ResponseThank you very much for spending time to comment our manuscript. It helps us to improve our manuscript significantly. Yes, you are right. Our manuscript just focus on the description of velvet ant and its host. However, at present, there are few articles address the parasitic biology of velvet ants, especially on bumblebee. Our manuscript contribute to understand the parasitic biology of velvet ant on bumblebee. It will be benefit to the bumblebee conservation. Therefore, we consider our manuscript fit to the journal area and hope to invoke the interesting of velvet ant and bumblebee researchers’ interesting. 

Comment 1The english language style should also be extensively corrected.

ResponseThanks. The manuscript was revised by Amerian Journal Experts company premier editor. The Certificate Verification Key: 6FA2-800E-C44D-790F-A6EP.

Comment 2More detailed comments in the attached file. 

Response: Many thanks for the detail comments. The comments help to improve our manuscript significantly. We response to the comments one by one. Please check the attached file. Thanks again for your improvement of our manuscript.

Reviewer 3 Report

Reviewer comment to the manuscript: insects-447872

Manuscript title: First record of the velvet ant Mutilla europaea(Hymenoptera: Mutillidae) parasitizing the bumblebee Bombus breviceps(Hymenoptera: Apidae) 

The authors describe a new host-parasite relationship between a velvet ant and a bumblebee species. This is a very important contribution to our knowledge because the biology of these wasps is now well understood and research on it is scarce. Therefore, the manuscript would definitely merit publication. However, for the manuscript to be fit for publication I think the authors need to address a few issues.

General comments

The abstract could give some more introductory information on mutilid wasps, such as how specialized they are as parasites and why it is interesting to investigate these parasites. Also, I would not really call this an experiment, because nothing was manipulated in the strict sense and compared to a control or different treatments. Furthermore, the study design of the observations was also not replicated. The introduction is well written and to the point. Material and methods are rather brief and should give some more detail, especially on the conducted observations. From reading point 2.3. I get the impression that the study was designed and conducted in a rather random fashion with no real goal in mind. It is not entirely clear how the observations were conducted and how data was recorded because the results are written in a rather anecdotal fashion and I get the impression that data was not recorded in an organized and structured way. It would be interesting to see some actual data, such as development times, the proportion of brood infected in a nest, how colonies became infected over time in the exposure study etc. I have to say, the pictures in the figures are excellent and definitely give a very good understanding of how the parasitizing of bumblebees works. Well done! The discussion is short but to the point.

Specific comments

Line 18: Should read: Mutilid wasps are…

Lines 64-76: Please add the information why on one hand you reared fresh colonies from wild queens and on the other hand why you collected established colonies from the wild.

Lines 79: Please add some information about what exactly was recorded during those observations and how observations were conducted. Or were you only looking for M. europaea?

Lines 84-94: Please indicate how many parasitized and clean colonies of B. breviceps were placed in the enclosure. Also please give some information about the reasons for putting B. haemorrhoidalis in the enclosure at all and why only two colonies. Also were the colonies provided with any food or was the enclosure providing the food, i.e. were there flowers. Please describe in more detail the enclosure.

Line 88: Please add how these “clean” colonies were arranged? Why did you choose these distances?

Line 92: How did you check if cocoons were infected? Can this be done without removal or damaging of the cocoon? Or did you remove the complete brood once you observed that the colonies were parasitized?

Lines 98: Please rephrase: 5 in 6 of the B. breviceps nests….

Line 113: What do you mean by “usually”? How were bumblebees parasitised in the unusual way?

Lines 122-128: How was development time measured? How many eggs/larvae were observed? What was the variation in developmental time? Did any of the parasitized bumblebee larva/pupae die? What was the prevalence of M. europaea exactly give numbers, please? All male pupae is a clear statement, but “few” is not. Probably make a figure for this.

Lines 130-135: A few more numbers on the spread of M. europaea would be great and how many nests were parasitized at a given time and distance. Connected to this, the prevalence of parasitized brood would be great to see in a table or graph.

Figures: The pictures are excellent! Thank you! They alone merit publication!

Table 1 Please add a legend to this table describing what is shown.

Author Response

 Reviewer # 3

The authors describe a new host-parasite relationship between a velvet ant and a bumblebee species. This is a very important contribution to our knowledge because the biology of these wasps is now well understood and research on it is scarce. Therefore, the manuscript would definitely merit publication. However, for the manuscript to be fit for publication I think the authors need to address a few issues.

Response: Thank you very much for spending time to comment our manuscript and confirm our work. Your comments improve our manuscript significantly. We have revised the manuscript according to your suggestion one by one. We hope the revised manuscript can reach the requirement of the journal.

Comments: General comments 

The abstract could give some more introductory information on mutilid wasps, such as how specialized they are as parasites and why it is interesting to investigate these parasites. Also, I would not really call this an experiment, because nothing was manipulated in the strict sense and compared to a control or different treatments. Furthermore, the study design of the observations was also not replicated. The introduction is well written and to the point. Material and methods are rather brief and should give some more detail, especially on the conducted observations. From reading point 2.3. I get the impression that the study was designed and conducted in a rather random fashion with no real goal in mind. It is not entirely clear how the observations were conducted and how data was recorded because the results are written in a rather anecdotal fashion and I get the impression that data was not recorded in an organized and structured way. It would be interesting to see some actual data, such as development times, the proportion of brood infected in a nest, how colonies became infected over time in the exposure study etc. I have to say, the pictures in the figures are excellent and definitely give a very good understanding of how the parasitizing of bumblebees works. Well done! The discussion is short but to the point.

 Response: Yes, you are right. Frankly speaking, our manuscript is the observation of velvet ant parasitizing on bumblebee. Therefore, the content is quite different to experimental manuscript. We consider that this manuscript fit to the journal and will be interested by velvet ant and bumblebee researchers.

Comment 1: Line 18: Should read: Mutilid wasps are…

Response: Thanks. We revised it.

Comment 2: Lines 64-76: Please add the information why on one hand you reared fresh colonies from wild queens and on the other hand why you collected established colonies from the wild.

ResponseRearing B. breviceps is the normal work in our lab. Sometimes, we collected the wild colonies to strengthen our lab bumblebee population in order to prevent the inbreed degeneration.

Comment 3: Lines 79: Please add some information about what exactly was recorded during those observations and how observations were conducted. Or were you only looking for M. europaea?

ResponseThank you for your suggestions. We are focusing on the M. europaea and the behavior of bumblebee was ignored. We revised the material and method part. Hope it express clearly this time.

Comment 4: Lines 84-94: Please indicate how many parasitized and clean colonies of B. breviceps were placed in the enclosure. Also please give some information about the reasons for putting B. haemorrhoidalis in the enclosure at all and why only two colonies. Also were the colonies provided with any food or was the enclosure providing the food, i.e. were there flowers. Please describe in more detail the enclosure.

Response: Thanks for the suggestions. We placed 5 parasitized colonies and 12 clean colonies of bumblebee in the yard. The clean colonies including 10 colonies of B. breviceps and 2 colonies of B. haemorrhoidalis. Because the number of B. haemorrhoidalis were small in our lab. All colonies were artificially fed with syrup and pollen. Actually, the colony can forage on wild flowers. We revised this part in manuscript.

Comment 5: Line 88: Please add how these “clean” colonies were arranged? Why did you choose these distances?

Response: We are sorry for the ambiguous expression. We revise this part. This is the first time that we observe the biology of velvet ant parasitizing on bumblebee. So, the choice of the distance was random. We didn’t know the dispersal ability of velvet ant. Even, it was not well design, we want to add this information. Hope it can help other researchers.

Comment 6: Line 92: How did you check if cocoons were infected? Can this be done without removal or damaging of the cocoon? Or did you remove the complete brood once you observed that the colonies were parasitized?

Response: We picked out the cocoon randomly and use scissors to cut the cocoon. Therefore, the cocoon was damage after the observation. At the end of the colony development, we remove the complete brood and count the parasitizing status. When we check the cocoon, we can see velvet ant female incubating the cocoon. We can find the cocoon was parasitized by velvet ant.

Comment 7: Lines 98: Please rephrase: 5 in 6 of the B. breviceps nests….

ResponseThanks. We rephrase the sentence.

Comment 8: Line 113: What do you mean by “usually”? How were bumblebees parasitised in the unusual way?

ResponseThanks. The expression is not exactly. We revised it.

Comment 9: Lines 122-128: How was development time measured? How many eggs/larvae were observed? What was the variation in developmental time? Did any of the parasitized bumblebee larva/pupae die? What was the prevalence of M. europaea exactly give numbers, please? All male pupae is a clear statement, but “few” is not. Probably make a figure for this.

ResponseWe are sorry for not supply the detail of development time. This experiment was done in 2016. At that time, we just measured six eggs and six larvae development time of velvet ant. It spend 26±1 days for from egg to adult. Other development stages was not recorded in detail. Yes, the bumblebee larvae died but didn’t rot. “prevalence” here we means the parasitize different bumblebee colonies. We revised the sentence to make it clearly.

Comment 10: Lines 130-135: A few more numbers on the spread of M. europaea would be great and how many nests were parasitized at a given time and distance. Connected to this, the prevalence of parasitized brood would be great to see in a table or graph.

Response: Yes, more detail about the spread of M. europaea. It will improve the quality significantly. We are sorry that we didn’t collect the data in 2016. This is the fault of our manuscript. We hope the description of velvet ant biology is interesting to researchers.

Comment 11: Table 1 Please add a legend to this table describing what is shown.

Response: Thanks. We added a caption to the table.

Round 2

Reviewer 2 Report

The majority of the comments were not directly addressed in the manuscript, but explained as a reply to the reviewer. These should have been addressed. The authors still did not include accurate results in terms of quantities and numbers and dismissed many of the suggestions not concerning the writing style. I believe that unfortunately at its present state, this manuscript is of very low quality and should not be published in a journal with the impact of Insects.

Author Response

Reviewer # 2

The manuscript presents a novel host association for velvet ants and bumblebees. There are serious flaws, however, with how the results are presented and the soundness of the discussion. The main flaw is that results are presented in a descriptive way, therefore making the article suitable for a more local, lower impact journal. There are no quantitative results and the discussion is highly speculative.

ResponseThank you very much for spending time to comment our manuscript. It helps us to improve our manuscript significantly. Yes, you are right. Our manuscript just focus on the description of velvet ant and its host. However, at present, there are few articles address the parasitic biology of velvet ants, especially on bumblebee. Our manuscript contribute to understand the parasitic biology of velvet ant on bumblebee. It will be benefit to the bumblebee conservation. Therefore, we consider our manuscript fit to the journal area and hope to invoke the interesting of velvet ant and bumblebee researchers’ interesting. 

Comment 1The english language style should also be extensively corrected.

ResponseThanks. The manuscript was revised by Amerian Journal Experts company premier editor. The Certificate Verification Key: 6FA2-800E-C44D-790F-A6EP.

Comment 2More detailed comments in the attached file. 

Response: Many thanks for the detail comments. The comments help to improve our manuscript significantly. We response to the comments one by one. Please check the attached file. Thanks again for your improvement of our manuscript.

From PDF:

Comment 1Line 24: This is a very broad statement given the limited distance tested. Please make it more specific based on your results.

ResponseThanks. We revised this sentence to make it more reality.

Comment 2Line 32: This is true for the whole family. I would write a first statement refering to ALL mutillids as "parasitoid wasps characterized by strong sexual dimorphism in which females are apterous and males are winged". Then introduce the particular genus.

ResponseThank for the advice. It may cause by different logic.

Comment 3Line 42: This statement seems out of nowhere. Any new host association would help understand it. I would remove it.

ResponseWe consider the most contribution of our manuscript is that we descript a new host of velvet ant.

Comment 4Line 47: I would be more specific here... 1840s sounds almost like 200 years ago

ResponseThanks. We added the exact time in this sentence.

Comment 5Line 54: Add author and year

ResponseThanks. We revised it.

Comment 6Line 58: I think you should be more specific with your aim. What does "shed further light" mean? This is too vague

ResponseThanks. We revised it.

Comment 7Line 76: Citing this figure here does not make sense. Do you mean figure 1C? Even then I think it would not make sense to refer to this figure here as it does not relate to transporation to the lab.

ResponseYes, it should be 1C. We want to should the colony that was collected from the wild.

Comment 8Line 78: if same as above specify.

ResponseThank you. we revised it. Hope it is more clearly.

Comment 9Line 81: How many? Please be more specific throughout your methods and results section, including numbers.

ResponseThanks. We revised all the relative sites throughout the manuscript.

Comment 10Line 91: for how long?

ResponseThanks. it means seven days. We revised it.

Comment 11Line 97: Why. Explain this sentence.

ResponseThere have some records for other bumblebee in this area. However, the population may affect by the climate very much. So, it is hard to find all the species.

Comment 12Line 99: This figure does not show nest location

ResponseThe figure is in a low resolution. We will replace it with high resolution. We can see some roots in the picture.

Comment 13Line 103: Table 1

ResponseThanks. We corrected it.

Comment 14Line 113: if not always, what else happened? Report numbers! Hoe many where parasitised at prepupal stage and how many not? This is a vague statement.

ResponseWe are sorry for the ambiguous. Only the prepupal stage of bumblebee were parasitized.

Comment 15Line 116: Again, why not report numbers? How many had 1,2,3,4 eggs? This is a very simple paper and to be up to the journal impact you need to report more specific results, otherwise it becomes a mere observation.

ResponseI am sorry that we didn't count the frequency of 1 to 4 eggs cocoon. However, we can see more one eggs from the pictures.

Comment 16Line 121: what happened to them? did they develop at all? How many?

ResponseWhen bumblebee pupae were parasitized. The pupae stop to develop. However, they didn't rot.

Comment 17Line 123: Again, you should report developmental times, means, etc.

ResponseThanks. We added it.

Comment 18Line 127: how many?

ResponseThanks. we revised it.

Comment 19Line 123: This is confusing. The beehives less than 20m away were supposed to also be mutillid-free, right? Please clarify these sentences as they are confusing.

ResponseAll the test colonies were mutillid-free except the five colonies with mutillid from the fields.

Comment 20Line 123: Why unfortunately? I think it is not appropriate here.

ResponseThanks. You are right. We removed it.

Comment 21Line 135: all after exactly one week? Again, report numbers.

ResponseSorry. We mean that when we see the adult of M. europaea in the beehive. We check the cocoon one week later and all of them were parasitized by M. europaea.

Comment 22Line 138: The figure is mislabeled. A does not show GPS points. I think A and B are switched around? Please fix

ResponseThanks. We revised it.

Comment 23Line 150: The table needs a legend.

ResponseThanks. We added it.

Comment 24Line 150

ResponseWe are sorry. It is hard to count the number during the field. We didn't have the exactly number of M. europaea, especially the big colonies.

Comment 25Line 159: compared to what? 80m is not enough to make this claim, otherwise support with more literature

ResponseThank you for the suggestion. We revised this sentence and make it clearly.

Comment 26Line 160: You found this in your results, if you mean it can use others than breviceps and haemorrhoidalis then the sentence needs to be reworded.

ResponseThanks. We want to express that M. europaea have many potential hosts.

Comment 27Line 161: This contradicts the "high vagility" claim.

ResponseWe mean the individual spread ability may lower. How they spread into large area was unknow. 

Comment 28Line 173: Explain further. How is this related to the previous evidence?

ResponseIt was observed in other bumblebees.

Comment 29Line 177: Support this with references

ResponseMany parasitizing insects like to parasitize on the male. For example, mites like to parasitize on the male cell of honeybee. So, we infer that they may be caused by the different nutrients.

Comment 30Line 178: The fact that only one out of three eggs can continue development does not have any relation with different nutrient availability between drones and females. It only means that drone nutrients are enough for just one egg. There is no evidence that nutrients from female or queen are not enough.

ResponseYes. You are right. This is just mean male pupae is enough for the development of velvet ant. We revised this sentence.

Comment 31Line 185: This is not necessarily true. Please provide dispersal rate for other mutillids and insects and compare.

ResponseYes, you are right. it should be compared. We concluded the result base on our field investigation and lab test. In the field, velvet ant was found around 3 km. Under the lab condition, they spread more than 20 m in seven days.

Comment 32Line 190: This could be questioned, given that only drones are affected and not entirely... this is why you need to specify how affected is drone development

ResponseYes, it will depend on the parasitizing male cocoon number. This is just inference. Our points are that when the male was affeced. The new virgin queen mating will be affected.

Comment 33Line 196: if they are immune, how would they affect them?

ResponseWe are sorry that we mean protect. How bumblebee protect themselves against velvet ant.

Reviewer 3 Report

The authors have addressed my comments to my satisfaction and I think the manuscript is of major interest to many researchers in the field. I have only a few little comments to add. Most of them are problems with language. The second sentence of the abstract seems wrong to me and should be revised. Line 93: something is missing in this sentence. Line 124 should read "these bumblebee pupae...". Line 130: this sentence sounds wrong please revise. Please add your response to my comments 6 and 9 to the manuscript. I think this information is of interest to the reader to know exactly how you removed brood and that these pupae were killed and how you measured developmental time.

Author Response

The authors have addressed my comments to my satisfaction and I think the manuscript is of major interest to many researchers in the field. I have only a few little comments to add. Most of them are problems with language.

Response: Thank you very much to satisfy with our revision and confirm the interest of many researchers.

Comment1: The second sentence of the abstract seems wrong to me and should be revised.

Response: Thanks. We rephrase the sentence.

Comment2: Line 93: something is missing in this sentence.

Response: Sorry for the confusing. We revised this sentence.

Comment3: Line 124 should read "these bumblebee pupae...".

Response: Thanks. We revised it.

Comment4: Line 130: this sentence sounds wrong please revise.

Response: Thanks. We rephrase the sentence.

Comment5: Please add your response to my comments 6 and 9 to the manuscript. I think this information is of interest to the reader to know exactly how you removed brood and that these pupae were killed and how you measured developmental time.

Response: Thanks. We have added these details and information to the Manuscript.

Round 3

Reviewer 2 Report

This version is largely similar to the version I received before, where comments were not addressed but answered to me. I add more comments and feel this needs work in order to be up to the impact of the journal.

Author Response

Response to Reviewers

Comments: This version is largely similar to the version I received before, where comments were not addressed but answered to me. I add more comments and feel this needs work in order to be up to the impact of the journal.

Response: Thank you very much for all the comments. We have tried our best to improve the manuscript according to the comments. We hope it can reach the impact of the journal.

Comment 1: Line 32: This is still not changed from the previous version. I believe it should be modified for accuracy and to avoid confusion.

Response: Thank you for the suggestion. We have revised the description to make it reasonable. The revised sentence is as following Most velvet ants (Hymenoptera: Mutillidae) are solitary parasitoid wasps characterized by strong sexual dimorphism in which females are apterous and males are winged. Mutilla is a genus of Mutillidae family”.

Comment 2: Line 42: Even though this is a contribution, this sentence should be removed as it is redundant to the previous one.

Response: Yes, it is redundant. We removed it. Thanks.

Comment 3: Line 47: No response or revision here?

ResponseThank you for the suggestion. We added the exact years and two references to clarify the vague expression. The revised sentence is as follows “The first description of Mutilla europaea L. within a nest of Bombus muscorum L. was provided by Christ in 1791 [reference 15], but the parasitic relationship between bumblebees and M. europaea was not confirmed until the 1840s [reference 4].”

Comment 4: Line 76: I would move the figure citation to The nests (Figure 1C) ... As it is placed now, the reference to the figure is not accurate.

Response: Thanks. We have changed the figure citation to Figure 1C.

Comment 5Line 97: I did not mean for you to explain it to me but to the reader. It is an open statement with no explanation.

Response: We revised the sentence to make it clear to readers. Thank you for your suggestion. The revised description is as follows “There have some records for other bumblebees in this area. However, we did not find nests of other species during the field investigation, because the population richness may affect by the climate very much”.

Comment 6: Line 135: This sentence needs to be re-written for clarity.

Response: Thank you for the suggestion. We revise the sentence. Hope it can clarify the meaning. The revised sentence is as follows “We sampled some cocoons randomly from each test colony and found all colonies were confirmed to contain M. europaea larvae one week after a female M. europaea entered the beehive.”

Comment 7Line 150: Please re-write to: Bumblebee B. breviceps and velvet ant M. europaea stages in each of the field colonies

Response: Thank very much for the revision. We have re-wrote the legend.

Comment 8Line 160: This sentence is still unclear. Please revise

Response: Thank you for the suggestion. We revised the sentence to make it clarity. The sentence is as follows “According to our investigation, M. europaea are high parasitizing in natural B. breviceps nests. It suggests that we should pay attention to the population dynamic changes of this bumblebee for conservation. In addition, the extensive bumblebee diversity in the area should also catch our attention as M. europaea can use different bumblebee species as hosts”.

Comment 9Line 173: Still, this sentence states that host aggression is affected by host status, and as worded sounds like a conclusion derived from the previous studies, but this is not demonstrated in either of the examples presented. If this is a separate conclusion drawn from a separate experiment then the sentence also needs re-wording.

Response: Yes, you are right. There is no evidence from our result. We didn’t found any fighting between velvet ant and bumblebee in our experiment. However, it was reported that they fight each other in bumblebee, Bombus ignitus. We have added reference to support our conclusion.

Comment 10Line 177: Yes, but you did not support your statement with any references. This needs at least reference and explain that in other species the males are prefered and what makes the males have more nutrient content.

Response: Thank very much for the suggestion. We have added a reference to support our inference.

Comment 11Line 185: Yes, but what makes you conclude this is a "rapid" dispersal? Rapid compared to what?

Response: Yes, we need a reference speed to compare in order to highlight dispersal speed. The revised sentence is as follows “The test of the dispersal ability in the laboratory also demonstrated that M. europaea is able to disperse at a relative high speed (about 80 m/20 days).”

Comment 12Line 190: Well then you need to explain this in the text, because many of your statements sound like "hand waving" with no support,

Response: When bumblebee colony was parasitized, the number of male cocoon will decrease and the number of male will down. In this study we found the parasitizing rate reach 100% at the end of colony life in our experiment. So, we concluded the male quality and quantity will be affected.